# Beyond Imaging and Genetic Signature in Glioblastoma: Radiogenomic Holistic Approach in Neuro-Oncology

**DOI:** 10.3390/biomedicines10123205

**Published:** 2022-12-09

**Authors:** Lidia Gatto, Enrico Franceschi, Alicia Tosoni, Vincenzo Di Nunno, Caterina Tonon, Raffaele Lodi, Raffaele Agati, Stefania Bartolini, Alba Ariela Brandes

**Affiliations:** 1Medical Oncology Department, Azienda USL of Bologna, 40139 Bologna, Italy; 2Nervous System Medical Oncology Department, IRCCS Istituto Delle Scienze Neurologiche di Bologna, 40139 Bologna, Italy; 3Department of Biomedical and Neuromotor Sciences, University of Bologna, 40139 Bologna, Italy; 4Functional and Molecular Neuroimaging Unit, IRCCS Istituto Delle Scienze Neurologiche di Bologna, 40139 Bologna, Italy; 5IRCCS Istituto Delle Scienze Neurologiche di Bologna, 40139 Bologna, Italy; 6Department of Neuroradiology, Bellaria Hospital, IRCCS Istituto Delle Scienze Neurologiche di Bologna, 40139 Bologna, Italy

**Keywords:** radiomics, radiogenomics, glioblastoma (GBM), diffusion weighted MR imaging (DWI), apparent diffusion coefficient (ADC), isocitrate dehydrogenase (IDH) mutation, O^6^-methylguanine-DNA methyl-transferase (MGMT) promoter methylation, pseudoprogression

## Abstract

Glioblastoma (GBM) is a malignant brain tumor exhibiting rapid and infiltrative growth, with less than 10% of patients surviving over 5 years, despite aggressive and multimodal treatments. The poor prognosis and the lack of effective pharmacological treatments are imputable to a remarkable histological and molecular heterogeneity of GBM, which has led, to date, to the failure of precision oncology and targeted therapies. Identification of molecular biomarkers is a paradigm for comprehensive and tailored treatments; nevertheless, biopsy sampling has proved to be invasive and limited. Radiogenomics is an emerging translational field of research aiming to study the correlation between radiographic signature and underlying gene expression. Although a research field still under development, not yet incorporated into routine clinical practice, it promises to be a useful non-invasive tool for future personalized/adaptive neuro-oncology. This review provides an up-to-date summary of the recent advancements in the use of magnetic resonance imaging (MRI) radiogenomics for the assessment of molecular markers of interest in GBM regarding prognosis and response to treatments, for monitoring recurrence, also providing insights into the potential efficacy of such an approach for survival prognostication. Despite a high sensitivity and specificity in almost all studies, accuracy, reproducibility and clinical value of radiomic features are the Achilles heel of this newborn tool. Looking into the future, investigators’ efforts should be directed towards standardization and a disciplined approach to data collection, algorithms, and statistical analysis.

## 1. Introduction

GBM is characterized by dismal prognosis, with a median survival of 12–15 months, high relapse rate, poor response to treatment, and morbidity [1,2,3].

GBM is a whole brain disease and not a focal malignant tumor, with diffuse and widespread infiltrative growth and clinically significant cellular proliferation also outside the tumor volume and into the surrounding brain tissue [4].

GBM stands for a group of diseases with remarkable genomic, histological, and imaging heterogeneity across patients and over time, leading to treatment resistance, tumor recurrence and progression. This complex picture poses several diagnostic and therapeutic challenges. This combined space–time histological and genetical heterogeneity is not only “intertumoral” but also within each patient, “intra-tumoral”, involves cell size, proliferation, metabolism, gene expression profile, phenotypic and epigenetic state, radiological phenotype, as well as vascularization and necrosis [5,6,7].

The current standard of treatment for GBM is the combination of maximal safe surgical resection, followed by temozolomid concurrent with and adjuvant to radiotherapy [2,8]. The heterogeneous phenotypic landscape of GBM makes this “uniform” standard of care inadequate and highlights the need for precision diagnostics, prognostication, and personalized treatment [9,10].

GBM was the first cancer to be sequenced by the cancer genome ATLAS (TCGA) research network [11], which identified the main genetic alterations and driver mutations that confer the overall complexity to the GBM genomic landscape. The initial publication reported mutually exclusive molecular alterations in three core pathways: the p53 pathway, the retinoblastoma (RB) pathway and the phosphatidylinositol 3-kinases (PI3K)/phosphatase and tensin homolog (PTEN) pathway. They found the p53 pathway was altered in 85% of GBMs, approximately 75% of tumors exhibited dysregulations in the RB pathway, and, finally, the PI3K/PTEN pathway disruption was described in 85–88% of GBMs [11].

Historically, GBM has been classified as isocitrate dehydrogenase (IDH) wild type and IDH-mutant (secondary GBM); nevertheless, according to 2021 WHO Classification of Tumors of the Central Nervous System [1], only grade 4 IDH wild type gliomas are now defined as GBMs. The presence of one or more of three genetic parameters, epidermal growth factor receptor (EGFR) amplification, the combination of gain of chromosome 7/loss of chromosome 10, and telomerase reverse transcriptase (TERT) promoter mutation is the molecular criterion for making the diagnosis of GBM, IDH wild type. Many of the studies reported in this review predate the latest WHO classification, therefore, they report the old nomenclature of “secondary” GBM, IDH mutant.

Approximately 40–50% of GBMs carry EGFR amplifications; among EGFR-amplified GBMs, in 20–50% of cases, a splice variant which creates a mutant form of EGFR (EGFR vIII) is present, conferring a more aggressive tumor biology [12,13].

O^6^-methylguanine-DNA methyl-transferase (MGMT) promoter methylation is present in approximately 35% of newly diagnosed GBM [14] and has been shown to be a predictive biomarker of response to DNA alkylating chemotherapeutics, such as temozolomide, since methylation can compromise the ability to repair DNA damage [15,16]. Notably, mutations of TERT promoter (a gene that controls telomeres, small portions of DNA found at the end of each chromosome, which protect the unstable DNA from degradation), have been reported in approximately 75–80% of GBM cases: its role as a prognostic/predictive factor is still uncertain and requires further study [15,16,17,18].

Clinical experience on GBM management demonstrates that precision medicine has failed in the treatment of brain tumors: trials of single-agent tyrosine kinase inhibitors have been uniformly disappointing and most studies with anti-EGFR agents have been negative. Additionally, predicting response to temozolomide is more complex than just determining the MGMT methylation status. It involves several aspects, including the percentage of methylation, its distribution within the diverse regions of tumor mass and the expression of several other key genes, such as EGFR VIII and p53 [19,20].

To date, the gold standard method for detecting these genomic biomarkers is histopathological examination, through immunohistochemistry and genomic sequencing. However, this method is invasive, expensive and time-consuming, and, due to intratumor heterogeneity, a small sample of tissue can increase the risk of erroneous genetic profiling.

In contrast, multi-parametric MRI is a powerful diagnostic alternative method that can facilitate the in vivo characterization of diverse aspects of the tumor and its micro-environment in a non-invasive and reproducible way [21,22].

P. Lambin coined the term “radiomics”, defining it as the process of extracting information from medical images, generally not appreciable with a mere visual examination, using advanced feature analysis [23]. Radiomics is, currently, an emerging, automated, high-throughput technique, which investigates how medical images can be transformed into quantitative data. Medical diagnostic imaging, in fact, produces an incredible amount of data, often underutilized for diagnosis, prognostication and research purposes [24]. The aim of radiomics is to extract a large number of quantitative parameters from medical images and correlate them with clinical or biological endpoints [25].

Radiomics, despite the current limitations, deriving from the low level of evidence of the studies and the heterogeneity of the methods used, might be useful to determine if baseline radiophenotype (of the primary tumor before treatment) is preserved in the follow-up scans after treatment, thereby evaluating the molecular profiling of recurrent tumors, the radiophenotypical variations over time, the longitudinal evolution of the mutational status, and treatment response [26]. It might be a useful tool in the field of precision oncology [27] to overcome the problem of tumors’ heterogeneity: unlike surgical biopsy, it assesses the whole three-dimensional tumor extent as well as the “tumor habitat”, the lesion margins, the surrounding peri-tumoral regions and the peri-tumoral edema sub-compartments [28].

It is likely that the greatest application of radiomics in neuro-oncology lies in radiogenomics, an active area of research investigating the relationship between quantitative features extracted from radiographic images and the respective underlying genomic pattern, to obtain tumor molecular characterization (e.g., gene expression profiles or mutations) on the basis of the tumor’s radiophenotype [23,29,30,31].

However, carefully checking the quality of the input data is challenging to guarantee a reproducible and robust output. The lack of standardization and the different methodologies adopted across diverse institutions makes studies’ validations challenging, representing a major obstacle in the translation of radiogenomics to clinical practice. Hence the need for larger prospective multicentric studies involving heterogenous populations [32].

This review provides a state-of-the-art description of the novel developments in the use of radiogenomics for the study of molecular markers of GBM and their potential for predicting recurrence and survival, particularly focusing on the applications of MRI radiomics.

## 2. Radiogenomics Workflow

Radiogenomic studies are designed following a systematic approach which includes several steps [33] (Figure 1): (1) image acquisition, (2) image pre-processing, (3) segmentation and identification of regions of interest, (4) feature extraction and quantification, (5) feature selection and reduction, (6) building of predictive and prognostic models using machine learning or deep learning, and (7) validation.

### 2.1. Image Acquisition

In this phase, a large pool of medical images of particular biological interest are acquired through various advanced MRI techniques, including diffusion weighted imaging (DWI), perfusion weighted imaging (PWI), and proton magnetic resonance spectroscopy (^1^H-MRS).

The quality of input data is of primary importance for the outcome of radiomic research and represents a challenge to ensure the reliability and reproducibility of model building: in future research, greater efforts should be directed towards a more precise and standardized data collection.

For example, Ellingson et al. [34], in a multicenter study investigating the quality of DWI data in GBM, showed that only 47% of patients had high quality data. The study evaluated the quality of DWI data using a five-point scoring system based on the following factors: (1) geometric distortion or artifacts on diffusion MR datasets; (2) apparent diffusion coefficient (ADC) values within white matter within an acceptable range of ~0.4–1.0 μm^2^/ms; and (3) ADC values within cerebrospinal fluid (CSF) within an acceptable range of ~2.5–4.0 μm^2^/ms. A five-point quantitative scaling scheme was used for each of these factors. The authors found that a total of 68% of patients had “usable” DWI data and only 47% of patients had high quality DWI data, concluding that the value of DWI data in multicenter trials was limited due to poor image quality.

The variability across institutions in the image acquisition step was a major issue for the collection of multi-center retrospective data for clinical trials on radiomics: variations across scanners [35], resolution, image reconstruction, slice thickness and contrast washout [36] were often limiting.

There are several important initiatives to standardize image acquisition across different institutions, as well as the Quantitative Imaging Network and the Quantitative Imaging Biomarkers Alliance® (QIBA) of Radiological Society of North America [37,38,39].

### 2.2. Pre-Processing of Data

Before feature extraction, the input data can be elaborated through a variety of preprocessing steps to improve image quality [33].

This is a phase of “normalization”, to delete possible “noise”, for example, by applying smoothing filters. It is crucial for harmonizing the input data in order to reduce statistical bias due to variability in image acquisition and different patient characteristics.

There are several pre-processing methods, including noise suppression, signal intensity normalization, bias field correction, image filtration, motion correction (to delete motion artifacts), voxel size standardization, signal dynamic range normalization and voxel intensity calibration [32,33,40]. Image filtration is used before the extraction of features to highlight particular image properties. Examples of image filters include Laplacian of Gaussian filters, which detect areas of rapid change (for example, edge) [33], and wavelet filters, which separate high- and low-spatial-frequency information.

### 2.3. Segmentation and Identification of Regions of Interest (ROI)

Segmentation involves the “volume of interest”, including the tumor region and subregions with distinct characteristics, reflecting the heterogeneity of cancer, named “tumor habitat”.

Quantitative analysis is performed only over the region of interest (ROI) that includes the tumor region and the tumor habitat, such as the lesion core, the margins of the lesion and the edema region. Thus, radiomics might be helpful for the microenvironment of the tumor, analyzing its heterogeneity [27]. The segmentation of ROI can be achieved by manual, semi-automated or completely automated methods (using deep learning algorithms) [32,41,42,43,44,45].

Manual segmentation of images is the most widely applied method but is time-consuming, operator-dependent and increases risk of inter-observer bias: if the ROI is too small, it cannot provide sufficient information, while if it is too extensive, it can cause an interpretation bias, due to the heterogeneity of the tumor. Therefore, manual or semi-automated segmentation should guarantee intra- and inter-observer reproducibility of the radiomic features and elimination of non-reproducible features from subsequent analyses.

In contrast, automatic segmentation is a privileged approach that ensures efficiency and reproducibility, but its success lies in the accuracy of the algorithm used [27].

Generalizability of algorithms is a major issue, and the application of the same algorithm on a different dataset often results in complete failure. Therefore, future research must be directed towards the application of reliable and reproducible algorithms for automated image segmentation [36,46].

To date, a universally accepted segmentation algorithm is not available: identifying criteria for the standardization of segmentation methods is a challenge for radiogenomics.

Among the various automated segmentation algorithms for brain tumor, some of the most notable are [47]:−Thresholding method: starting from a grayscale image, thresholding returns a binary image [48];−Clustering: a more elaborate procedure that allows the determination, starting from a set of data, of groups with “similar” characteristics;−Edge-based method: emphasizes areas of abrupt change within a digital image (for example, discontinuity in the physical properties of tissues), which, generally, reflect changes in the physical status of the tissues [49];−Region growing: a simple region-based segmentation method, based on the selection of pixels that are similar and, therefore, can be classified as appertaining to the same tumoral subregion [50];−Watershed algorithm: a unique segmentation tool where gray levels and voxels are classified by their intensity or gradient in a topographical map, with ridges and valleys;−Atlas method: a tumor-free reference MRI is used to contour the MRI image containing the tumor volume [51].

There are many commercial software solutions usually used for brain segmentation: FMRIB Software Library (FSL), Statistical Parametric Mapping (SPM), and Brainsuite are the most common [52]; with regard to deep learning models, U-Net [53,54] is progressively imposing.

FSL [52] is a software created by members of the Analysis Group, FMRIB, Oxford, U.K. (URL: http://www.fmrib.ox.ac.uk/fsl/, accessed on 8 December 2022). Segmentation in FSL takes place through two different steps:The first step consists of BET (brain extraction tool). It is a procedure where a first segmentation, which includes brain tissue and beyond, is performed. All structures which do not contain only brain tissue and which can cause biases (eyes, muscle, base of neck, scalp, fat, cerebrospinal fluid) are eliminated with a completely automatic algorithm;The second step consists of FAST (FMRIB’s automated segmentation tool), that is, the segmentation of the brain volume previously extracted with the BET. FAST is a package, included in the FSL software, for segmentation of the brain volume into the three different tissues (gray matter, white matter and CSF, the latter exclusively contained within the volume extracted with the BET), including algorithms for spatial intensity corrections (also called bias fields).SPM software (current version: SPM12) [52] uses a tool named optimized voxel-based morphometry which was developed at the Institute of Neurology at University College of London (UCL Queen’s Square Institute of Neurology, Queen’s Square House, Queen’s Square, London, WC1N 3BG, UK) and is available from the web.

Brainsuite is a collection of image analysis tools including:−Tools for brain surface extraction, bias field correction, voxel classification, cerebrum labeling, and surface generation;−Tools for processing of diffusion data including tensor fitting and tractography;−Sophisticated tools for visualizing and exploring MRI data, diffusion data, tractography and connectivity.

### 2.4. Feature Extraction

After image segmentation and processing, the extraction of radiomic features that quantitatively describe the patterns of oncological phenotypes, can finally be performed. This step is critical, as it implies the extraction of high-dimensional features that are processed by specific software (for example, PyRadiomics v3.0.1, an open-source solution for the extraction of radiomics data from medical images [46,55], Computational Environment for Radiological Research [56], or Imaging Biomarker Explorer [57,58,59]). The image biomarker standardization initiative (IBSI) is an independent international working group that collaborates to standardize the extraction of radiomic images, providing consensus-based guidelines [60].

Quantitative features are classified into the following groups [32]:(a)Texture features, describing the variation of gray level values within the tumor;(b)Shape features, describing form and geometrical properties of the region of interest, such as surface, volume, compactness, diameter and sphericity;(c)Histogram-based features, calculated starting from the histogram that describes the distribution of pixels in the ROI, the mean, median, maximum, minimum values of the voxel intensities on the image, asymmetry, kurtosis (flatness), uniformity, and entropy;(d)Second-order features derived from the gray-level co-occurrence matrix, quantifying the incidence of voxels with same intensity;(e)Higher order features: features that describe the relationships between two or more pixels of the ROI, obtained after applying filters (e.g., wavelet transform, Laplacian transform, Gaussian filter, etc.) or mathematical transform to the pictures [61].

### 2.5. Methods for Dimensionality Reduction and Feature Selection

Based on the software used, many of the extracted features are redundant. In this phase, it is important to focus on “dimensionality reduction” and feature selection for generating valid and generalizable results. To achieve this, redundant and “weak” features must be removed from the model; this selection reduces the possibility of overfitting. However, the issue of interpretability of features after dimensionality reduction should be considered [53].

The most common methods for dimensionality reduction include cluster analysis and principal component analysis (PCA), which use linear transformations of the input features), kernel PCA, and autoencoders (which use nonlinear transformations) [33].

PCA aims to create a smaller set of representative variables called principal components from a large set of features, organizing a group of maximally uncorrelated variables from a large set of correlated variables. The output of PCA is represented by score plots, that provide a graphical instrument to classify elements in the data sets for similarity [61].

The most popular feature selection methods include recursive feature selection, least absolute shrinkage and selection operator (Lasso) and variance thresholding. Lasso feature selection is a widely used regression analysis method that performs feature selection, removing useless, redundant or noninformative features, making the statistical model easier, simple, with fewer parameters, thus, increasing the prediction value of the model [33].

### 2.6. Classification of Radiomic Features and Informatic Analysis: Machine Learning and Deep Learning

Once a subset of top features correlated with the expected outcome is identified, machine learning classifiers and different statistical methods are used to build predictive and prognostic models [32,62].

Machine learning is the field of study that gives computers the ability to learn without being programmed. Informatic analysis of radiomic features usually involves two main categories: classic machine learning (such as support vector machine (SVM) and random forest [63,64]), and deep learning methods using convolutional neural networks (CNNs) that, in most recent years, has taken over the field, outperforming classic machine learning methods (Figure 2) [45,65].

Machine learning is a branch of artificial intelligence that studies algorithms capable of learning from data, synthesizing new knowledge from them. It can improve the knowledge of the system to be studied by observing the input data (training phase) and then providing outputs such as prediction or decision making, by comparing these patterns to new unseen data to carry out a certain output.

Classic machine learning techniques use hand-engineered features, putting them into statistical tools such as SVM and random forest to organize and segregate the data (Figure 2) [66].

Random forest is a statistical model for classification, regression and other tasks, that outperforms a multitude of decision trees from a subset of data. It may require more data than SVM [61,67]. SVM is a supervised learning technique that analyzes data for both classification and regression analysis and outperforms a set of hyperplanes in a high dimensional space to segregate the data into two classes in the feature space. It is useful in the case of high-dimensional input data, but interpretation of the final model can be difficult [33].

Machine learning uses different learning prediction models including supervised and unsupervised approaches [68,69].

In supervised learning, the endpoint, such as tumor response or tumor grading, is known, and the algorithm uses a large amount of training inputs in order to learn a rule that connects inputs to their corresponding outputs [70,71,72].

Unsupervised learning, instead, allows for a more exploratory approach: the final outcome is unknown and the input data are unlabeled, thus, the algorithm is trained to identify unknown patterns, hence the name “unsupervised.”

The main limit to accuracy of ML models is represented by overfitting and underfitting.

Overfitting occurs when the population of the dataset is low compared with the number of features that describe each subject. The extraction of high-dimensional, large-scale features captures noise, thus, the developed model achieves good performance within the training set, resulting, instead, as unsuitable for the validation data.

Underfitting, in contrast, due to small sample sizes of the features within the model, occurs when the algorithms fail to capture certain patterns of the input data that are informative. It can be overcome by techniques such as synthetic minority over sampling technique (SMOTE) [33]. To improve the output and overcome the overfitting, normalization methods are needed. Skull stripping is a preliminary processing technique to separate the brain tissue from other tissues, which are a major obstacle for automatic brain image segmentation and analysis techniques [47,73].

Early studies adopting ML are mostly single institution, using small datasets and heterogeneous methods for segmentation. This leads to lack of interpretability and standardization. To overcome the issue of imaging heterogeneity across sites and institutions, the Medical Image Computing and Computer Assisted Intervention Society introduced an annual Brain Tumor Segmentation Challenge in 2012 [74]. The Brain Tumor Segmentation (BraTS) is a multi-institutional dataset of manually segmented pre-operative multiparametric MRI scans containing high- grade and low-grade gliomas imaging acquired with different protocols, that has evolved over the years, with a continuously increasing number of patient cases [74]. The BraTS protocol consists of structural MRI volumes: (a) a native T1-weighted scan (T1), (b) a post-contrast T1-weighted scan (T1Gd), (c) a native T2-weighted scan (T2), and (d) a T2 fluid attenuated inversion recovery (T2-FLAIR) (T1, T1Gd, T2, T2-FLAIR) with delineations of the relevant tumor sub-regions.

The last BraTS update, dated 2018, focused not only on the segmentation to brain tumor sub-regions, but also included clinical data such as patient age, overall survival, and resection status, to facilitate the second objective, which is to predict survival.

Multiple studies have explored the value of deep learning (DL) approaches for predicting prognosis, tumor grades and molecular profiles in GBM, and for distinguishing progressive disease from pseudoprogression after treatment [75,76].

DL is the preferred method when a large amount of data is included in the cohort.

The term DL refers to algorithms hierarchically organized on multiple levels, hundreds of layers of neural networks (hence the term “deep”), that automatically extract critical features from a subset of data (Figure 2) [70,71,72,77,78]. Since 2014, the use of CNN models, actually considered as state-of-the-art for segmentation, continues to increase, achieving excellent results [66,70,79,80,81,82,83].

A CNN consists of two orders of layers: the convolution layers and the pooling layers. The units in the convolution layers are organized into feature maps, in which each layer is connected to the next one through a convolution layer that finds local conjunctions between features of the previous layer. The role of the pooling layer is to semantically merge similar features into one. Once the convolution process is complete, there is a layer that introduces non-linearity to the model, increasing its complexity.

The advantages of deep learning approaches are listed below:−It is not necessary to segment the tumor;−It is not necessary to explicitly define the features to be calculated;−It is not necessary to select the features.

The limits of deep learning, instead, are:
−Larger input data are needed;−Problems of interpretability.

### 2.7. ROC Curve and Model Validation

Basically, statistical analysis is required to identify the features that may be related to the molecular signature of GBM and to eliminate the redundant features. Then, the selected features are uploaded into a machine learning model for predicting molecular profiles [67].

Reproducibility and clinical value of the model are estimated in the validation step, first tested with internal cross-validation and then validated on independent external cohorts [61].

The predictive performance of the model is measured using the area below the receiver operating characteristic (ROC) curve (AUC). The ROC curve is an analysis of the diagnostic performance of a clinical test. It is a statistical technique that measures the accuracy of a diagnostic test along the entire range of possible values. The area below the curve is a number between 0 and 1. If this measure is closer to 1, the model is more specific.

Notably, in machine learning and deep learning algorithms, the segmentation performance is often measured by the Dice score, which is a measure of how similar the objects are. The Dice coefficient is a measure of overlap between two masks that ranges from 0 to 1; 1 indicates a perfect overlap, while 0 indicates no overlap. The Dice coefficient should not be greater than 1 [47].

## 3. Radiogenomics of Glioblastoma

Radiogenomics is a powerful method for studying the biology of GBM, which has demonstrated its ability to characterize GBM, to predict molecular signature (in particular the status of the MGMT promoter methylation and IDH mutation) and to determine therapeutic response and survival of newly diagnosed patients.

Almost all studies have demonstrated a good level of accuracy, with very high sensitivity and specificity, but there is a need to standardize the methods and algorithms for the computation of radiomic features.

Most of the literature within radiogenomics concerns neuro-oncology [84]. In 2008, Diehn et al. combined radiogenomics with microarray DNA analysis in order to noninvasively map the gene expression within the tumor [85]. The study highlighted a strong association between morphological aspects of GBM captured with radiomics and gene expression, confirming the hypothesis that neuroimaging reflects the underlying gene-expression pattern. Particularly, genes involved in angiogenesis and tumor hypoxia (e.g., VEGF, SERPINE1, PLAUR) resulted as strongly correlated with the contrast enhancement phenotype. Similarly, a robust association between the genes involved in proliferation and cell-cycle progression (e.g., TOP2A, CDC2, and BUB1B) and a specific mass effect radiological phenotype, was observed. In addition, a high C:N ratio (the ratio of the contrast-enhancing volume to the necrotic tumor volume) resulted as correlating with overexpression of the gene EGFR [85].

In a study analyzing microRNA, gene expression and quantitative MR-imaging data in GBM, a high expression of periostin (POSTN)—associated with a worse prognosis and poor survival in GBM patients—and low expression of miR-219, were found in the most aggressive subtype, the mesenchymal subtype [86]. POSTN is a determinant of cellular invasion and GBM aggressivity, and miR-219 is regulator of cellular invasion by binding to the 39UTR of the POSTN gene, thus, decreasing POSTN protein levels. The authors identified distinct radiomic features capable of accurately predicting periostin overexpression in GBM patients.

Zinn et al. performed a radiomic textural analysis on a dataset of 29 TCGA GBM patients to investigate a possible relationship between the three most frequent driver mutations (p53, PTEN, EGFR) promoting GBM proliferation and imaging characteristics. Interestingly, they described distinct “radiomic profiles” associated with the classical pattern of genetical alterations of GBM, p53, PTEN, and EGFR [87].

Hu et al. performed an interesting and unique study, where multiparametric MRI and texture analyses were matched with the genetic status of several subregions of the tumor, by collecting 48 image-guided biopsies from 13 GBMs. The study demonstrated significant imaging correlations (univariate analysis) for six driver genes: EGFR, platelet-derived growth factor receptors (PDGFR), PTEN, cyclin-dependent kinase inhibitor 2A (CDKN2A), RB, and p53 [88]. Interestingly, the authors observed that within a single GBM tumor, distinct regional genetic subtypes may coexist.

Radiogenomics research, progressively, has moved away from broad-range genetic analyses and has, subsequently, focused on the use of imaging features for specific molecular subtype prediction. IDH 1 and 2 mutations and MGMT methylation status are biomarkers widely used in clinical practice due to their high predictive and prognostic value [89], and have received relevant attention in GBM radiogenomic research.

### 3.1. Prediction of IDH Mutational Status

The mutant IDH status plays an important role in gliomagenesis and is an independent, well-known prognostic and predictive biomarker in patients with gliomas, having significant implications in terms of increased overall survival and chemo-sensitivity.

Mutations of the IDH gene family lead to accumulation of the oncometabolite 2-hydroxyglutarate (2-HG), that confers characteristics of less aggressiveness to tumor cells when compared with IDH wild-type tumors [90].

Currently, immunohistochemical staining and DNA sequencing are the most common methods for determining the IDH mutational status in gliomas; therefore, several radiogenomic studies have suggested that radiophenotypic appearance of GBM can non-invasively provide direct insight into the molecular signature.

The most relevant radiogenomic studies predicting IDH mutation are summarized in Table 1.

Several radiogenomic studies have suggested connections between IDH mutation and tumor location, reporting that, overall, IDH mutations seem to occur, more frequently, in the frontal lobe [29,91,103,104].

Tejadaa Nejra et al. [91] performed a large prospective study on 237 patients with newly diagnosed GBM, and 131 patients with lower-grade glioma, aimed at establishing any elective tumor locations in relation to GBM genotype [91]. The segmentation was performed on MRI images with a semi-automated approach through a voxel-based lesion symptom mapping (VLSM) analysis. They observed a concordant predilection for the frontal lobe location, adjacent to the rostral extension of the lateral ventricles in IDH mutant gliomas cohorts (GBMs and low-grade gliomas). Furthermore, a large region of no enhancing tumor, cysts with low T1, suppressed T2-FLAIR signal intensity and a higher ratio of the T2-weighted to T1-weighted contrast enhanced volumes, were described as features predictive of IDH mutant status [92,93,103].

A large radiomic retrospective multicenter study, aimed at predicting IDH mutation status through a random forest classification, extracted 1614 imaging features from 225 GBM patients [94]. Four single-region radiomics models were built from tumor core, whole tumor, peritumoral edema region and other tumor regions. The model combining all-region radiomic features with a clinical parameter and age, by using SMOTE algorithm, achieved the best accuracy (97%) [94].

In the last 5 years, the use of DWI, diffusion tensor imaging (DTI), arterial spin labeling (ASL) perfusion MRI imaging and MRI spectroscopy, which provide further tumoral pathophysiology information, is progressively growing and becoming promising for the prediction of IDH mutation status in gliomas [94,105,106].

DWI uses the diffusion of water molecules to generate contrast in MR images and allows researchers to study how water molecules diffuse through tissues.

Recent radiomic studies of diffusion-based MRI indicate significant differences for minimum or mean ADC values in the enhancing regions of the tumor for IDH-mutant tumors compared with the wild-type counterpart.

Xing et al. performed a retrospective study analyzing DWI, DSC-PWI, and conventional MR imaging in 42 patients with a diagnosis of grade II and III astrocytoma. They found that minimum ADC was significantly higher in IDH-mutated tumors than in IDH wild-type counterpart. They established a threshold value of ≥1.01 × 10^−3^ mm^2^/s, able to discriminate the two groups (IDH mutated and IDH wild type) with a sensitivity and specificity, respectively, of 77% and 82%. A combination, instead, of conventional MR imaging, DWI, and DSC-PWI techniques provided a relevant predicting value, resulting in a sensitivity and specificity of 92% and 91% [95].

A retrospective study performed on 176 GBM patients conducted by Hong et al. demonstrated that a higher proportion of insular involvement, a larger tumor volume, a higher enhancing portion on the contrast-enhanced T1 sequences, a higher ratio between T2-weighted to T1-weighted contrast-enhanced volumes and a higher ADC, were strongly associated with IDH mutation [93].

A large study by Wu et al., including 131 patients with diffuse gliomas, both LGG and GBM, correlated MRI phenotype and ADC not only with molecular markers (IDH mutation, 1p/19q codeletion status, MGMT methylation) but also with tumor “aggressiveness” and survival. IDH wild-type gliomas tended to exhibit a lower mean relative ADC (*p* < 0.001) than IDH-mutant gliomas. In addition, they found that a lower mean relative ADC was strongly associated with poor survival in both IDH mutant and IDH wild-type tumors, regardless of grading and genotype [96].

Water distribution within human tissues usually follows a Gaussian curve. However, the heterogeneity of the tissues can modify the diffusion of the water molecules, making it chaotic, and leading to non-Gaussian diffusion.

DTI can reflect the anisotropic diffusion of water in vivo. Diffusion kurtosis imaging (DKI), an extension of the DTI, can provide more precise information on tissue characteristics by quantifying the degree of deviation from the Gaussian curve. The parameters that can be derived from DKI are the mean diffusivity (MD), fractional anisotropy (FA), mean kurtosis (MK), kurtosis fractional anisotropy (KFA), and mean kurtosis tensor (MKT) [107].

Alis et al. analyzed 142 patients with a diagnosis of high-grade glioma, demonstrating that kurtosis plays significant role in IDH status determination [108].

This result was confirmed by Bisdas et al., who, by an SVM analysis, demonstrated that kurtosis is a reliable measure in IDH genotype prediction, with an accuracy of 81% [109].

Other machine learning studies have reported interesting results in predicting IDH status integrating multimodal MRI patterns with clinical data.

Zhang et al. [97] performed a machine-learning based retrospective study on 120 patients with primary grade III (*n* = 35) and IV (*n* = 85) gliomas to predict the IDH status in HGG, integrating clinical data (such as age, sex, Karnofsky performance status, and pre-operative steroid use) with MRI features. The most predictive features (both clinical and radiological) resulted as age, frontal or temporal tumor location, ADC, laterality, andT2/FLAIR volume. Patient age, particularly, resulted as the most important clinical variable in the model. This is not surprising, if we consider that patients with IDH-mutated tumors are younger than the IDH wild-type counterpart. The study achieved an accuracy, in the prediction of IDH genotype in high-grade gliomas, respectively, of 86% in the training data set and 89% in the data set. The model combining clinical features with MRI data achieved the best performance in the prediction of IDH genotype, with accuracies of 77.78% and 85.17% in the training set and in the validation set, respectively.

Similarly, in the study by Zhou et al. [110], histogram, shape, and texture features were extracted from T1-contrast-enhanced and T2-FLAIR images of preoperative MRIs of 538 glioma patients, and correlated with age in order to predict IDH mutation status using a random forest algorithm. This model achieved a high AUC (0.92 and 0.91, respectively, for the training and the validation set).

Similar to DWI, multiple PWI–MRI studies have been assessed for IDH genotype prediction in GBM, with relative cerebral blood volume (rCBV) which reflects tumor vascularity, being the parameter most frequently employed [75]. This suggests that tumor angiogenesis and vessels distribution are different in IDH-mutant gliomas compared with the wild-type counterpart, and these differences may be distinguishable based on DSC perfusion MRI patterns [98].

However, while ADC measurements reproducibility is well recognized, dynamic susceptibility contrast (DSC) MRI workflow is not standardized and suffers from several biases derived from variable protocols applied across different institutions [75,111,112].

In a cohort study of 73 glioma patients, Kickingereder and colleagues [99] demonstrated that IDH-mutated gliomas were characterized by lower rCBV relative to their wild-type counterpart. Using a histogram for rCBV values, each unit increase was associated with a decrease in the likelihood of IDH mutation. They reported quite good performance of this method in predicting IDH status, with specificity and sensibility of 89% and 78%, respectively. In addition, the study confirmed the significant inhibition of hypoxia-inducible-factor 1-alpha (HIF1A) in IDH-mutated tumors.

Yamashita et al. [100] suggested that absolute tumor blood flow (derived from the cerebral blood flow maps of arterial spin labeling imaging), relative tumor blood flow, necrosis area, and percentage of cross-sectional (necrosis area inside the enhancing lesion) were significantly higher in IDH wild-type tumors than in the mutant counterpart. The performance in predicting IDH genotype, evaluated by ROC analysis, resulted as acceptable (AUC was, respectively, 0.850 for absolute tumor blood flow, 0.873 for relative tumor blood flow, and 0.739 for necrosis area).

Sudre et al. performed a multicenter study to determine the diagnostic value of machine learning assisted DSC-MRI techniques for classifying glioma grade and IDH genotype, using a random forest algorithm. They found a lower rCBV in IDH-mutated tumors and performed a reliable stratification of patients by IDH genotype using DSC-MRI extracted perfusion texture features and shape features [101].

Overall, the multimodal combination of CBV and ADC seemed to lead to better results for predicting IDH status and GBM aggressiveness [75].

However, the real promising revolution in radiogenomics seems to be the application of deep learning to brain tumors.

An interesting deep learning study was conducted on 259 patients from a TCIA set, with either low- or high-grade gliomas, to predict IDH mutation status, 1p/19q codeletion, and MGMT promoter methylation status [92]. The researchers used a pretrained algorithm for tumor segmentation and PCA to extract the clusters of meaningful features for successful classification. The features predictive of IDH-mutant status resulted as in line with the existing literature: presence of a larger portion of non-enhancing tumor, central necrotic cystic areas with low T1 and FLAIR suppression, and well-defined tumor margins. IDH wild-type tumors, instead, tended to demonstrate a larger portion of enhancing tumor with peripheral enhancement and an infiltrative pattern of edema. In this study, no cross-validation with external dataset was performed.

Bangalore Yogananda et al. [102] reported their own fully automated MRI-based deep learning model to assess IDH mutational status. They examined 214 patients affected by gliomas from a TCIA set by a fully automated network performing tumor segmentation and IDH status prediction simultaneously, based on 3D MRI images (3D-Dense-UNets approach) They achieved an accuracy of 97.14%, specificity of 98%, and sensitivity of 98% in predicting IDH genotype. They also demonstrated that IDH classification using only T2-weighted images had comparable performance if compared with a multi-contrast network. One possible explanation was that deep learning networks using conventional single-mode MRI images reduced the effect of head movement, allowing much shorter image acquisition times. A limit of this study was that cross-validation with external testing on a separate dataset was not performed.

In 2019, Choi et al. conducted a retrospective study enrolling 463 patients affected by glioma (grades II-IV) to classify IDH mutational status using a deep learning application for DSC perfusion MRI, named recurrent neural network (deep learning model that learns sequential patterns or temporal dependencies within time-series data). They reported interesting results in IDH genotype predictions, achieving an accuracy, sensitivity, and specificity, respectively, of 92%, 92%, and 93% in the validation set (AUC = 0.96 for GBM cohort) [98].

In 2021, a Korean study reviewed 1166 preoperative MR images of WHO grade II-IV gliomas, including both IDH wild type or IDH mutant, to non-invasively predict the IDH genotype from preoperative MR images, using a fully automated approach with CNNs. Their deep learning model, a CNN-based classifier using 2D and 3D tumor images demonstrated an accuracy of 93.8%, proving to be a highly reliable tool for the noninvasive prediction of the IDH status [113].

Given the unique biology value of 2-HG, its detection by magnetic resonance spectroscopy (MRS) would be a valid tool for the assessment of IDH genotype [114]. Branzoli et al. confirmed the value of MRS in glioma patients for the detection of 2-HG; in addition, they compared the performance of the two most used 2-HG MRS techniques (the long echo modulation and the J-difference spectral editing using the Mescher–Garwood scheme), concluding that the latter exhibited a superior level of accuracy [115].

In conclusion, despite the difficulty in comparing the results of the various studies as they were very inhomogeneous, the most relevant findings regarding radiogenomics performance in IDH mutational status prediction can be summarized as follows (Figure 3):−Gliomas harboring IDH mutations occurred, more frequently, in the frontal lobe, adjacent to the rostral extension of the lateral ventricles;−A larger tumor volume in T2 sequences and a higher volume ratio between T2 and T1 sequences with contrast agents were observed in IDH mutant tumors, together with the presence of a high portion of non-enhancing tumor and a central necrotic cystic area with low T1 and FLAIR suppression. A larger portion of enhancing tumor with peripheral enhancement and an infiltrative pattern of edema, instead, was strongly associated with IDH wild-type genotype;−In diffusion imaging, a higher mean ADC value was observed in IDH mutated tumors;−Tumor vascularity, neoangiogenesis and vessels distribution, reflected by the parameter rCBV, were much less represented in IDH-mutated tumors than in the wild-type counterpart. Consequently, IDH-mutated gliomas exhibited significantly lower rCBV values relative to their wild-type counterpart;−Higher skewness and kurtosis were associated with IDH mutational status;−Approaches based on multimodal combination of CBV and ADC seemed to lead to better results for predicting IDH status and GBM aggressiveness [76];−Radiomics models combining data from multiple tumor regions, for example, core, whole tumor and peritumoral edema region, were more accurate in IDH prediction, especially if the analysis was integrated with clinical data (age, performance status, surgery);−ML-based approaches integrating clinical data (mainly age, significantly lower in IDH-mutated tumors) with the most predictive radiological features (frontal tumor location, ADC andT2/FLAIR volume) achieved the best accuracy in the prediction of IDH genotype;−DL approaches using DSC perfusion MRI images accurately predicted the IDH mutational status [76];−Approaches based on 2-HG MRS techniques also achieved adequate accuracy, sensitivity, and specificity in the prediction of the IDH status.

Despite good performance in IDH prediction, both CNN and radiomics have major challenges in clinical practice translation: the first obstacle is tumor segmentation, as manual segmentation is time-consuming and automatic segmentation has still poor reproducibility. The second obstacle is the lack of a standard method for the selection and computation of features. Third, even if CNN might eliminate the issue of feature computation and selection, procedures and models are not standardized, as the studies performed to date are very inhomogeneous.

Furthermore, all the radiomic/CNN studies performed to date have been retrospective, and prospective trials are lacking: this is also a critical barrier to clinical translation.

### 3.2. Prediction of MGMT Promoter Methylation Status

MGMT is a gene encoding for a DNA repair protein, crucial for genomic stability.

Temozolomide acts in damaging the DNA, generating mutant DNA containing O^6^-methylguanine, that leads to cell death. This modification, usually, is effectively repaired by the MGMT protein that reverses the effect of chemotherapy by restoring purine from O^6^-methylguanine [116]. When the MGMT promoter is silenced through methylation, the MGMT protein is expressed at lower levels and DNA repair cannot be performed.

It is accepted that the methylation status of the MGMT promoter is a favorable prognostic factor in patients with GBM, associated with a more robust response to alkylating agents such as temozolomide, higher response to radiotherapy and longer survival [117,118,119,120].

However, MGMT promoter methylation, which is present in approximately 40–50% of the cases, is not ubiquitous, being area specific, and may change over time during the disease course, between primary tumor and recurrence [121,122,123]. This implies that a single biopsy specimen may be not representative of the entire tumor mass and, therefore, may direct the clinician towards incorrect therapeutic strategies.

Prediction of MGMT methylation status, based on MRI, seems to be challenging, as it provides a non-invasive diagnostic methodology for patient stratification and treatment planning.

Relevant radiogenomic studies predicting MGMT methylation status are summarized in Table 2.

Previous non-radiomic studies introduced the role of imaging characteristics in MRI, such as tumor necrosis, enhancement patterns and tumor location, for the prediction of MGMT methylation status. Kanas et al. [129] reported that MGMT unmethylated tumors tended to exhibit more homogenous contrast enhancement, while MGMT methylated tumors were characterized by ring contrast enhancement, with central necrosis and decreased peritumoral edema. Other studies reported that MGMT unmethylated GBM was located in the right frontal lobe or in proximity to the SVZ.

Korfiatis et al. [124] performed a large radiomic retrospective study enrolling 155 GBM patients with known MGMT methylation status, and compared several different classes of texture features, showing that the combination of four texture features (correlation, energy, entropy, and local intensity) provided a really valid potential tool for the prediction of the MGMT methylation status in GBM. Their SVM-based algorithm achieved about 80% sensitivity and specificity in the prediction of MGMT methylation status. These results, subsequently confirmed by other studies [130,131], exemplified how ML models can be used to non-invasively obtain information on MGMT methylation status in preoperative GBM.

Several radiogenomic studies support the worth of DWI as a possible surrogate method to assess the MGMT methylation status in GBM [125,132], with higher ADC values reported in methylated GBMs relative to their unmethylated counterparts.

Moon et al. [125] reported that MGMT methylated GBMs exhibited higher ADC values than the unmethylated group (*p* = 0.055). In contrast, the rCBV ratio was not different between the two groups (*p* = 0.380).

In a retrospective study analyzing 108 GBM patients, an intratumoral subregion both with high T1 contrast enhancement and low ADC, named “high risk volume (HRV)”, was identified on multi-parametric MRI, predicting both unmethylated MGMT status and shorter survival (*p* < 0.001 and *p* = 0.038, respectively, in the discovery and validation cohort) [133].

Rundle-Thiele et al. [134] were the first to raise the question of the method used for the analysis of diffusivity measures. Assuming that ADC ability to predict MGMT status has shown mixed results, they explored if, within the same patient cohort, the prediction of the MGMT status may be subject to change based on the method selected to analyze ADC measures. They reported a retrospective analysis of 32 patients with GBM with MGMT status already known. The used two diverse methods to measure ADC: the minimum ADC, and a two-mixture model histogram approach. They observed a strong relationship between an elevated “minimum ADC” and methylation status. In contrast, using the two-mixture normal distribution histogram analysis, they found that the mean ADC was significantly lower in the methylated MGMT patient group than in the unmethylated patient (*p* < 0.0246). This study emphasized how the method selected to analyze ADC measures significantly influences the prediction of MGMT status.

Multi-habitat MRI radiomics is, currently, emerging as a valuable method for the prediction of MGMT methylation status and prognosis of GBM patients: Wei et al. [126] introduced a comprehensive model integrating radiomic features, clinical variables and two ADC values (the tumor and edema areas) for determining MGMT methylation status.

They concluded that radiomic features extracted from T1-contrast and T2-FLAIR sequences performed higher than those extracted from the ADC sequence, probably because of the relatively poor imaging resolution of ADC, that limited the stability and robustness of the derived radiomic features [126].

A further interesting observation was that MGMT methylated GBMs exhibited higher rCBV [135] in studies involving DSC and arterial spin labeling [127].

In 2018, a meta-analysis about the value of radiomics for MGMT status prediction demonstrated an overall sensitivity and specificity of 79% and 73%, respectively—a not particularly encouraging result [136]. These data suggest that radiogenomics is still insufficient for use in the clinical setting, and far from being employed as a common tool for the detection of MGMT methylation status.

Only a few more recent machine learning and deep learning studies have achieved better results. Haiianfar et al. [137] performed a study on 82 patients affected by GBM, aimed at non-invasively predicting the MGMT gene promoter status by using MRI radiomics features. Tumors were manually segmented in four regions: (1) whole tumor, (2) active/enhanced region, (3) necrotic regions, and (4) edema regions. The edema region resulted as the top-performing region in the prediction of MGMT status using multivariate analysis (AUC 0.78); the inverse variance feature from gray level co-occurrence matrix in whole tumor, instead, had the best performance using univariate analysis (*p*-value = 0.002). Chen et al. [128] proposed a deep learning model using contrast-enhanced T1W images and FLAIR images for the prediction of MGMT status, with encouraging results. The study enrolled 87 GBM patients; FLAIR images resulted as the best predictor of MGMT status (Dice score = 0.897).

In conclusion, several radiogenomic studies have assessed the MGMT methylation status in GBM. Despite the difficulty in comparing the results of the various studies as they were very inhomogeneous, the most relevant findings can be briefly summarized as follows (Figure 3):−MGMT methylated tumors were localized in the left hemisphere, especially in the left temporal lobe. In contrast, MGMT unmethylated tumors tended to be localized in the right hemisphere, in the right frontal lobe or in proximity to the SVZ;−MGMT unmethylated tumors tended to exhibit more homogenous contrast enhancement, while MGMT methylated tumors were characterized by ring contrast enhancement, with central necrosis and decreased peritumoral edema;−In T2/FLAIR images, MGMT methylated tumors had a lower hyperintense tumor volume, in contrast with unmethylated tumors;−In diffusion imaging, increased minimum ADC values and higher ADC ratio were associated with MGMT promoter methylation;−In perfusion imaging, higher rCBV was associated with MGMT promoter methylation;−Multi-habitat MRI and comprehensive multi-omics models integrating radiomic features (possibly from both the tumor and the edema areas), clinical variables, and genetic data achieved the best accuracy for determining MGMT methylation status [128].

### 3.3. Discrimination of Pseudoprogression from Early Progression

The standard protocol of treatment in newly diagnosed GBM consists of temozolomide concurrent with and adjuvant to radiotherapy [2]. During the first 6 months of follow-up, nearly 20% to 30% of patients experience pseudoprogression, a condition in which the size of the tumor often increases and/or new inflammatory lesions appear at MRI, simulating disease progression in the absence, however, of neurological clinical signs of deterioration or worsening. These lesions generally tend to stabilize over time and stop growing further [116,138,139].

Discrimination of pseudoprogression from early progression is a real challenge in neuro-oncology practice. If pseudoprogression is suspected, temozolomide should be continued with close radiological follow-up. Thus, the final diagnosis in these patients can only be made retrospectively—if they improve without a change of second line therapy.

The Response Assessment in Neuro-Oncology (RANO) criteria are used as an alternative to surgical biopsy for distinguishing pseudoprogression from true progression, but with limited and variable diagnostic value.

In this context, radiomics can improve the diagnostic performance, particularly when combined with information on MGMT promoter status.

A recent retrospective study of 76 patients (53 early progressions and 23 pseudoprogressions) performed using 11 radiomics features, tried to discriminate between early progression and pseudoprogression, especially when combined with MGMT promoter status. The study achieved good sensitivity but poor specificity (81.6% sensitivity, 50.0% specificity, in training phase), with a moderate performance improvement after combining the data with information regarding the methylation status [140].

Elshafeey et al. confirmed that pseudoprogressive and progressive disease exhibit distinct radiomic features that could be extracted by MR perfusion parameters analysis combined with SVM [141]. They used both kurtosis and rCBV-based maps, selecting the top 220 features (60 kurtosis features and 160 rCBV features) that achieved the highest predictive accuracy in differentiating between progression and pseudoprogression, and subsequently built a radiomic model using SVM that achieved AUC 94%, sensitivity 92%, and specificity 100% in discriminatory power. The key features selected for both kurtosis and rCBV models were entropy, sum of squares, and autocorrelation.

Kim et al. [142] developed a radiomics model to differentiate pseudoprogression from early tumor progression using multiparametric MRI, in particular, extracting radiomic features from contrast-enhanced T1-FLAIR imaging, as well as ADC and CBV maps. A large study on 105 GBM patients demonstrated that combining 3D shape and surface radiomic features extracted from the lesion habitat (T1WI enhancing lesion and T2WI/FLAIR hyperintense perilesional region) could capture differences between real progression and pseudoprogression with high accuracy (90%) [143].

In general, the multiparametric radiomics showed higher performance in the external validation and internal validation than any single approach (ADC or CBV parameter) [142].

Qian et al. were the first to introduce the radiogenomics approach to detect candidate genes for pseudoprogression in GBM, identifying interferon regulatory factor (IRF9) and X-ray repair cross-complementing gene (XRCC1) as potential biomarkers of pseudoprogression [144].

### 3.4. Survival Prognostication

Over 40% of GBM patients do not respond to standard radio-chemotherapy treatment and develop disease progression within a few months, on average 6–9 months. Hypoxia in GBM is a key pathway known to promote tumor neovascularization, cell proliferation and treatment resistance [32].

Beig et al. [145] were the first to propose a very unique radiomic study, constructing a hypoxia enrichment score (HES) to predict the extent of hypoxia and survival of GBM patients. Analyzing the data from data from 85 GBM patients, they constructed a radiomic risk score (RRS) using radiomic features from different tumor habitats to stratify GBM patients according to their survival. Additionally, they provided a biological basis for the RRS, identifying 192 genes exhibiting a different expression profile between the “high-risk” and “low-risk” groups using gene ontology and single-sample gene set enrichment analysis. Statistically significant correlations (*p* < 0.05) were found between the shape features of the peri-tumoral edema region (i.e., sphericity, elongation and convexity) and biological processes of cell proliferation and neovascularization. The extracted features strongly associated with HES could also distinguish between short-term survivors (OS < 7 months) and long-term survivors (OS > 16 months) (*p* = 0.003) (Table 3) [145].

In 2014, a study [146] examined the correlation between GBM OS and morphologic imaging features and hemodynamic parameters obtained from the non-enhancing region of the tumor, along with clinical and genomic markers. Poor OS (*p* = 0.0103) and PFS (*p* = 0.0223) were associated with increasing values of rCBV of the non-enhancing region of the tumor. The EGFR wild-type genotype associated with high rCBV in the non-enhancing region of the tumor exhibited the worst prognosis (AUC 0.62) (Table 3). Regarding clinical and imaging presurgical prognostic factors, rCBV of the non-enhancing region of the tumor resulted as the top predictor; also important were the Karnofsky performance status, age at diagnosis, and non-enhancing region crossing the midline.

Choi et al. [147] evaluated the potential of radiomics when combined with conventional clinical and genetic prognostic models for improving OS and PFS prognostication in GBM patients. A total of 120 patients were included in this retrospective single-center study. The prognostic performances resulted as improved when radiomics was added to the clinical model (AUC for OS improved from 0.62 to 0.73; AUC for PFS improved from 0.58 to 0.66), genetic model (AUC for OS improved from 0.59 to 0.67; AUC for PFS improved from 0.59 to 0.65), and combined model (AUC for OS improved from 0.65 to 0.73; AUC for PFS improved from 0.62 to 0.67) (Table 3).

Kazerooni et al. [148] assessed the additional value of integration of multi-omics data (clinical, radiomic and genetic data) for the accurate prediction of survival and clinical outcome of patients with newly diagnosed GBM. They built multiple models starting from the base model, including basic clinical data such as age, gender, and extent of resection, and showed incremental performance to predict OS in GBM patients by adding multiple layers of prognostic information, including radiomics, MGMT methylation, and genomic data obtained by NGS sequencing of the tumor samples (Table 3).

## 4. Conclusions

Radiogenomics represents an emerging field that aims at improving the results provided by radiology and genomics: medical images are more than pictures, they are data useful for predicting the genomic profile of tumors [30].

GBM, because of its heterogeneity, genetic instability and complicated assessment of treatment response, is the ideal candidate for radiogenomics and represents an issue where this methodical approach can contribute the best of its potential.

However, with the improvement of radiogenomics, all its limitations are rising to the surface: standardizing the guidelines for systematic image acquisition and segmentation algorithms is challenging.

With the incremental inclusion of artificial intelligence [149], large-scale data sharing, and CNNs combined with clinical data, neuroradiologists’ performances are likely to improve, allowing more precise, impactful diagnoses.

Nevertheless, the major challenges limiting radiomic approaches are the poor reproducibility of studies, and the variability and lack of consistency attributed to the absence of standardized procedures. Furthermore, radiomics studies are mostly retrospective, thus, having a low level of evidence. Therefore, despite the great potential of the radiogenomic approach, it currently does not find application in clinical practice, remaining confined only to the field of research.

In the future, multi-institutional prospective clinical trials should be devoted to improving reproducibility and applicability of radiogenomic protocols, in order to focus on the concept of personalized/adaptive medicine, instead of outdated precision medicine.

## Figures and Tables

**Figure 1 biomedicines-10-03205-f001:**
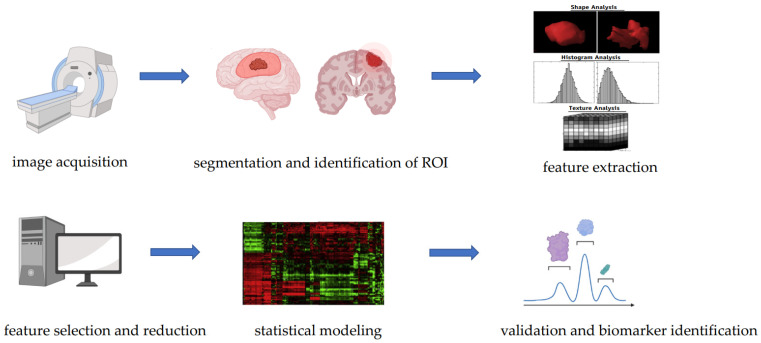
Steps of a radiogenomic study.

**Figure 2 biomedicines-10-03205-f002:**
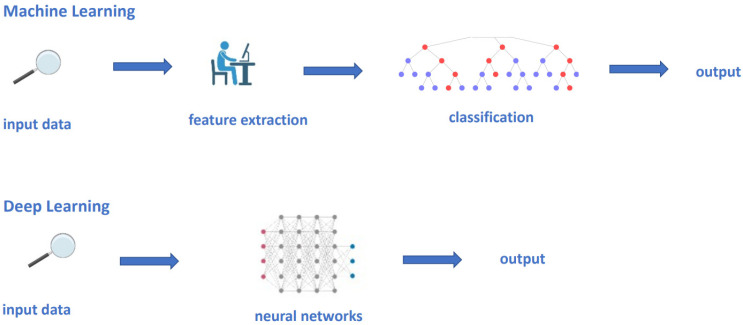
Conventional machine learning and deep learning models. In a conventional machine learning process, an expert need to define the meaningful features to be processed into the statistical model, which will work out the output, based upon the selected features. Contrastingly, in deep learning models, the manual definition of features is not needed; each network hierarchy automatically extracts critical features from a subset of data.

**Figure 3 biomedicines-10-03205-f003:**
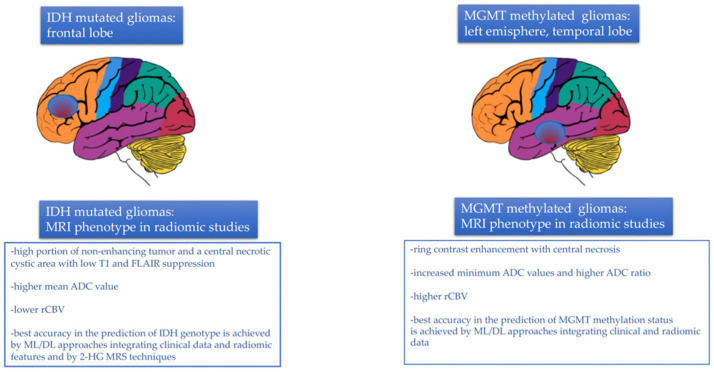
MRI tumor phenotype in relation to IDH mutational status and MGMT methylation status assessed by radiomic studies.

**Table 1 biomedicines-10-03205-t001:** Relevant radiogenomic studies predicting IDH mutational status.

Study	MRI Technique	IDH1 Mutant MRI Phenotype/Predicitive Features	Number of Patients	Performance AUC/Accuracy Value
Tejadaa Nejra et al., 2018 [91]	VLSM analysis of MRI images	Frontal lobe location, adjacent to the rostral extension of the lateral ventricles	237	Permutation-adjusted *p*-value = 0.021
Chang et al., 2018 [92]	T2, FLAIR, and T1 pre and postcontrast	Absent or minimal areas of enhancement, central areas of cysts with low T1 and FLAIR suppression, and well-defined tumor margins	259	94% accuracy
Hong et al., 2018 [93]	T2 and T1CE and DWI	Larger volume on T2 and a higher volume ratio between T2 and T1CE; higher mean ADC	176	AUC was 0.48 for T2 volume; 0.73 for T2-T1 volume ratio; 0.65 for ADC mean
Li et al., 2018 [94]	T1, T1CE, T2 and FLAIR	The multiregional model built with all-region features performed better than the single-region models, while combining age with all-region features achieved the best performance	225	AUC 0.96
Xing et al., 2017 [95]	DWI, DSC-PWI and conventional MRI imaging	Minimum ADC and relative ADC significantly higher; relative maximum CBV <2.35 predictive of IDH mutation	42	AUC was 0.87 for minimum ADC, 0.84 for relative ADC and 0.82 for relative maximum CBV
Wu et al., 2018 [96]	Conventional MRI imaging	Higher enhancement, necrosis and edema, and a higher mean relative ADC	131	AUC 0.79
Zhang et al., 2016 [97]	Machine learning algorithm to generate a model predictive of IDH genotype based on the integration of clinical features and conventional MRI features (Statistics and Machine Learning Toolbox MATLAB 2015a)	Top features resulted were age and MRI parametric intensity, texture, and shape features	120	AUC 0.92
Choi et al., 2019 [98]	T1, T2, T2-FLAIR, T1CE, DSC perfusion MRI	The recurrent neural network model (RNN) accurately predicted the IDH status using DSC perfusion MRI	463	AUC 0.96 for GBM patients
Kickingereder et al., 2015 [99]	T1 images both before and after administration of gadoterate meglumine (Dotarem, Guerbet) as well as axial FLAIR and axial T2 images	Lower rCBV	181	92.2% accuracy
Yamashita et al., 2015 [100]	T1CE, precontrast T1 spin-echo, T2-turbo spin-echo, FLAIR and DWI	Higher absolute tumor blood flow, relative tumor blood flow, necrosis area, and percentage of cross-sectional necrosis area inside the enhancing lesion. No significant difference in the ADC minimum and ADC mean	66	AUC for absolute tumor blood flow, relative tumor blood flow, percentage of cross-sectional necrosis area inside the enhancing lesion, and necrosis area were 0.850, 0.873, 0.739, and 0.772, respectively
Sudre et al., 2020 [101]	Machine learning assisted DSC-MRI using random forest classifier	Lower tumor surface to volume ratio (SAV) and measure of non-compactness; higher skewness and kurtosis; higher correlation and sum entropy	333	Overall specificity of 77% and sensitivity of 65%
Bangalore Yogananda et al., 2019 [102]	MRI-based deep learning 3D-Dense-UNets	High IDH classification accuracy of T2w image-only network (T2-net)	214	T2-net demonstrated AUC of 0.98 ± 0.01

ADC = apparent diffusion coefficient; DWI = diffusion-weighted MR imaging; CE = contrast enhancement; IDH = isocitrate dehydrogenase; MGMT = O^6^-methylguanine-DNA methyl-transferase; DSC-PWI = dynamic susceptibility contrast-enhanced perfusion-weighted imaging; CBF = cerebral blood flow; CBV = cerebral blood volume; rCBV = relative cerebral blood volume; rCBF = relative cerebral blood flow; RNN = deep learning model that learns sequential patterns or temporal dependencies within time-series data.

**Table 2 biomedicines-10-03205-t002:** Relevant radiogenomic studies predicting MGMT methylation status.

Study	MRI Technique	MGMT Methylated Tumors MRI Phenotype/Predicitive Features	Number of Patients	Performance AUC/Accuracy Value
Chang et al., 2018 [92]	T1, T1CE, T2, T2 FLAIR	Heterogeneous, nodular enhancement; presence of an eccentric cyst; edema with cortical involvement; frontal and superficial temporal predominance	259 patients	Accuracy 83%
Korfiatis et al., 2016 [124]	T2-fast spin-echo, axial T1 and T1CE. Two supervised machine-learning classifiers were used to predict MGMT methylation status: SVM-based classifier and random forest	The best-performing classification system resulted from SVM with features extracted from T2 images	155	AUC 0.85
Moon et al., 2012 [125]	Axial T1, axial T2-fast spin-echo sequence, axial FLAIR, axial T2-gradient-echo sequence	Higher ADC value and higher ADC ratio in the methylated group; rCBV ratio did not differ between the two groups	38	ADC values tended to be higher in the methylated group. ADC ratio was significantly higher in the methylated group. rCBV ratio did not differ between the two groups (*p* = 0.380)
Wei et al., 2019 [126]	T1CE, T2 FLAIR and DWI	A fusion radiomics signature combining four single radiomics signatures (T1-WI-tumor, T1-WI-edema, T2-FLAIR-tumor, and T2-FLAIR-edema) showed optimal performance in predicting the MGMT methylation status	105	AUC of 0.925 in the training cohort and 0.902 in the validation cohort
Han et al., 2018 [127]	Diffusion-weighted (DWI) and 3-diminsional pseudo-continuous arterial spin labeling (3D pCASL) imaging	MGMT promoter methylation was associated with tumor location and necrosis (*p* < 0.05). Increased ADC value (*p* < 0.001) and decreased rCBF (*p* < 0.001) were associated with MGMT promoter methylation. ADC achieved better predicting efficacy than rCBF (ADC: AUC, 0.860; vs. rCBF: AUC, 0.835) The combination of tumor location, necrosis, ADC and rCBF resulted in the highest performance in predicting the MGMT promoter methylation	92	The combination of tumor location, necrosis, ADC and rCBF resulted in the highest AUC of 0.914
Chen et al., 2020 [128]	Deep learning model analyzing contrast-enhanced T1images, FLAIR images	FLAIR images showed the better tumor segmentation performance and the better MGMT status prediction	106 patients	Accuracy = 0.827 ± 0.056

ADC = apparent diffusion coefficient; DWI = diffusion weighted MR imaging; MGMT = O^6^-methylguanine-DNA methyl-transferase; DSC-PWI = dynamic susceptibility contrast-enhanced perfusion-weighted imaging, CBF = cerebral blood flow; rCBV = relative cerebral blood volume; rCBF = relative cerebral blood flow.

**Table 3 biomedicines-10-03205-t003:** GBM survival prognostication.

Study	MRI Technique	Survival Prognostication	Number of Patients	Performance AUC/Accuracy Value
Beig et al. [145]	T1, T2, T2 FLAIR	Use of 25 radiomic features from the tumor habitat predicted PFS	203	*p* < 0.0001 on the training set and *p* = 0.03 on the test set
Jain et al. [146]	Dynamic susceptibility contrast-enhanced T2-weighted perfusion MR imaging	Worsening OS and PFS were associated with increasing relative cerebral blood volume obtained from the non-enhancing region of GBM	45	OS (*p* = 0.0103); PFS (*p* = 0.0223)
Choi et al. [147]	T2, T2 FLAIR, T1CE	Radiomics added to the clinical model achieved the best performance in PFS and OS prognostication	120	AUC = 0.66 for PFS AUC = 0.73 for OS
Kazerooni et al. [148]	Pre-operative MRI acquisition on a 3 Tesla scanner	Multi-omics data (clinical, radiomic and genetic data) achieved better performance in predicting OS	516	AUC = 0.78 in the discovery cohort AUC = 0.75 in the replication cohort

## Data Availability

Not applicable.

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
