# Peer review of "Beyond Imaging and Genetic Signature in Glioblastoma: Radiogenomic Holistic Approach in Neuro-Oncology"

_biomedicines, 2022, doi:10.3390/biomedicines10123205_

Round 1

Reviewer 1 Report

This manuscript provides a comprehensive review on the development of the use of radiogenomic in neuro-oncology field. Although this technique is immature, it is worth the exploring.

There are some minor concerns: 1) Figure 1 and 2 need figure legends to briefly introduce the figures.

2) The  description of fig 1 and 2 provided too much technical details, the reader may not be interesting. It will be better if write it briefly.

3) In the sections of " prediction of IDH mutation status" and " prediction of MGMT promoter methylation status". Although the authors listed lots of reports about the current data and development. The reader still are confused with the data. It will be much better if the authors summarize the characters of radiogenomic that may be  corresponding to the status of  IDH mutation and the status of mgmt promoter methylation.

Author Response

Rev#1

  • There are some minor concerns: 1) Figure 1 and 2 need figure legends to briefly introduce the figures.

Author response:

We thank the reviewer for this comment. We have introduced figure legend at the end of the manuscript

  • The description of fig 1 and 2 provided too much technical details, the reader may not be interesting. It will be better if write it briefly.

 Author response:

We thank the reviewer for this comment. We have shortened the technical details of the captions.

  • In the sections of " prediction of IDH mutation status" and " prediction of MGMT promoter methylation status". Although the authors listed lots of reports about the current data and development. The reader still are confused with the data. It will be much better if the authors summarize the characters of radiogenomic that may be corresponding to the status of IDH mutation and the status of mgmt promoter methylation.

 Author response:

We thank the reviewer for this comment.  We have discussed common findings and discrepancies of the studies regarding the status of IDH mutation and the status of MGMT promoter methylation, summarizing the most relevant findings. You can find it in yellow on pages 28, 29 and 33.

Reviewer 2 Report

Review for biomedicines-2038076

In the present manuscript, the authors review the recent advancements in the use of radiogenomics for the assessment of molecular markers of interest in GBM regarding prognosis and response to treatments, and monitoring recurrence. This knowledge may provide insights into the potential efficacy of such an approach for survival prognostication.

In general, the submitted manuscript is well structured. The abstract clearly explains the objective of the review and describes the significant contribution of this review to the field. The introduction provides enough theoretical considerations of the different topics, however, the first 2 thirds are difficult to follow. The “conducting wire” between ideas should be improved to provide a good integration of the knowledge.

Some inconsistencies should be addressed:

General:

a)     There are some paragraphs with missing literature references (e.g. page 3).

b)     Please revise the abbreviations. Some are not written in cursive. Some are written in cursive many times. Some are not written in cursive the first time the abbreviation appears. Also, I suggest standardizing all the abbreviations (e.g. avoid LASSO and lasso).

c)     Please consider reducing the number of paragraphs. In case of paragraphs addressing the same subject, they should be merged into only one paragraph (e.g. pages 1, 8, 9, etc.). It will help the reader to better understand the text. I’m not referring to bullet points.

d)     Please consider to revise the repetitions of definitions and state them all in the introductory and “RADIOGENOMICS WORKFLOW“ parts. It is clear for the reader that different people wrote different parts, since possibly different co-authors felt the need of providing the same definition. This should not be perceived in a manuscript.

e)     Information in the summary tables should be harmonized and the text simplified whenever possible.

Page 3 - The authors only focus on MRI scans. However, the scope of the concept of radiomics is broader, as medical images from CT, PET, SPECT, etc. can also be used. This should be addressed by the authors in the introduction before focusing only on MRI scans.

Page 3 – It would be interesting to improve Figure 1 to address all the 7 topics covered in the manuscript, from image acquisition to model validation.

Page 6 – I suggest including BET and FAST explanations within the respective bullet point. It will help the reader.

Page 6 – Why Brainsuite was not detailed as it was done for FSL and SPM.

Page 8 – The paragraph “An example for radiogenomics application…” seems to appear out of context. The English of this text should also be revised.

Page 15 – Please revise the following paragraph “However, while ADC measurements reproducibility is well recognized, dynamic susceptibility contrast DSC-MRI dynamic susceptibility contrast (DSC)-MRIs workflow (…)”.

Page 21 – Which are the differentiating radiomic features shown by Elshafey et al.? These are not detailed.

The small conclusions provided by the authors at the end of each topic are a very positive point to highlight. However, it would be great if authors could provide a summary image with a clear association between predictive factor (e.g. IDH mutational status, MGMT promoter methylation status, etc..) and the respective most promissing radiomic features reviewed.

Overall, this review is worth of publication in Biomedicines with major reviews.

Author Response

Rev#2

  • There are some paragraphs with missing literature references (e.g. page 3).

Author response: We thank the reviewer for this comment. We have introduced literature references where missing. You can find it in the text in yellow on pages 5-7 (see ref 11, 25, 26, 28, 29)

  • Please revise the abbreviations. Some are not written in cursive. Some are written in cursive many times. Some are not written in cursive the first time the abbreviation appears. Also, I suggest standardizing all the abbreviations (e.g. avoid LASSO and lasso).

Author response: We thank the reviewer for this comment. We have revised the abbreviations.

  • Please consider reducing the number of paragraphs. In case of paragraphs addressing the same subject, they should be merged into only one paragraph (e.g. pages 1, 8, 9, etc.). It will help the reader to better understand the text. I’m not referring to bullet points.

Author response: We thank the reviewer for this comment. We have reduced the number of paragraphs (only 4 paragraphs)

  • Please consider to revise the repetitions of definitions and state them all in the introductory and “RADIOGENOMICS WORKFLOW“ parts. It is clear for the reader that different people wrote different parts, since possibly different co-authors felt the need of providing the same definition. This should not be perceived in a manuscript. 

Author response: We thank the reviewer for this comment. We have revised the definitions included in the manuscript and harmonized the different parts.

  • Information in the summary tables should be harmonized and the text simplified whenever possible

Author response: We thank the reviewer for this comment. We have simplified and harmonized the tables.

  • The authors only focus on MRI scans. However, the scope of the concept of radiomics is broader, as medical images from CT, PET, SPECT, etc. can also be used. This should be addressed by the authors in the introduction before focusing only on MRI scans. 

Author response: We thank the reviewer for this comment. We have explained that this review is focused on MRI radiogenomic approach both in the abstract and in the introduction. You can find it in yellow on pages 2 and 8.

  • It would be interesting to improve Figure 1 to address all the 7 topics covered in the manuscript, from image acquisition to model validation.

Author response: We thank the reviewer for this comment. We have improved Figure 1 addressing all the topics covered in the manuscript

  • I suggest including BET and FAST explanations within the respective bullet point. It will help the reader.

Author response: We thank the reviewer for this comment.  We have included BET and FAST explanation within the respective bullet points.

  • Why Brainsuite was not detailed as it was done for FSL and SPM.

Author response: We thank the reviewer for this comment.  We have explained Brainsuite. You can find it in yellow on page 12.

  • The paragraph “An example for radiogenomics application…” seems to appear out of context. The English of this text should also be revised.

Author response: We thank the reviewer for this comment.  We have removed the paragraph as it is superfluous and difficult to understand.

  • Please revise the following paragraph “However, while ADC measurements reproducibility is well recognized, dynamic susceptibility contrast DSC-MRI dynamic susceptibility contrast (DSC)-MRIs workflow (…)”.

Author response: We thank the reviewer for this comment.  We have revised the paragraph.

  • Which are the differentiating radiomic features shown by Elshafey et al.? These are not detailed. 

Author response: We thank the reviewer for this comment. We have explained and detailed

the differentiating radiomic features shown by Elshafeey et al. You can find it in yellow on page 34.

  • The small conclusions provided by the authors at the end of each topic are a very positive point to highlight. However, it would be great if authors could provide a summary image with a clear association between predictive factor (e.g. IDH mutational status, MGMT promoter methylation status, etc..) and the respective most promissing radiomic features reviewed.

Author response: We thank the reviewer for this comment.  We have discussed common findings and discrepancies of the studies regarding the status of IDH mutation and the status of MGMT promoter methylation, summarizing the most relevant findings. You can find it in yellow on pages 28, 29 and 33.

Reviewer 3 Report

In their paper titled: "Beyond imaging and genetic signature in glioblastoma: radiogenomic holistic approach in neuro-oncology", the authors report on the state of the art in the application of machine learning to the identification and classification of glioblastomas. The paper is precisely elaborated and logically organised. Only a few minor comments are appropriate here:
1. There are missing captions next to the images. I found them at the end of the manuscript, but they should be in an appropriate place in the text below the figures.
2. there is a typo on p.15 in the section: "...259 patients from a TCIA set, with either....", please correct this.

Author Response

Rev#3

  • There are missing captions next to the images. I found them at the end of the manuscript, but they should be in an appropriate place in the text below the figures.

Author response: Thank you for this comment. We have introduced captions below the figures.

  • there is a typo on p.15 in the section: "...259 patients from a TCIA set, with either....", please correct this.

Author response: Thank you for this comment. We have correct the wrong word “promotor”, replacing it with “promoter”.

Round 2

Reviewer 2 Report

2nd Review for biomedicines-2038076

Several considerations were not carefully addressed. I highlight below some of those.

  • Please consider to revise the repetitions of definitions and state them all in the introductory and “RADIOGENOMICS WORKFLOW“ parts. It is clear for the reader that different people wrote different parts, since possibly different co-authors felt the need of providing the same definition. This should not be perceived in a manuscript. 

Author response: We thank the reviewer for this comment. We have revised the definitions included in the manuscript and harmonized the different parts.

Reviewer response: There are still other sections of the manuscript needing this revision. E.g. Authors revised page 6 and 17, but lots of repeated texts are seen.

  • The authors only focus on MRI scans. However, the scope of the concept of radiomics is broader, as medical images from CT, PET, SPECT, etc. can also be used. This should be addressed by the authors in the introduction before focusing only on MRI scans. 

Author response: We thank the reviewer for this comment. We have explained that this review is focused on MRI radiogenomic approach both in the abstract and in the introduction. You can find it in yellow on pages 2 and 8.

Reviewer response: Pages 8 does not refer this.

  • The small conclusions provided by the authors at the end of each topic are a very positive point to highlight. However, it would be great if authors could provide a summary image with a clear association between predictive factor (e.g. IDH mutational status, MGMT promoter methylation status, etc..) and the respective most promissing radiomic features reviewed.

Author response: We thank the reviewer for this comment.  We have discussed common findings and discrepancies of the studies regarding the status of IDH mutation and the status of MGMT promoter methylation, summarizing the most relevant findings. You can find it in yellow on pages 28, 29 and 33.

Reviewer response: Pages 28, 29 and 33 are references. The texts added throughout the text are very confused, repetitive…please clarify if the intention was to put it using bullet points.

Author Response

  • Please consider to revise the repetitions of definitions and state them all in the introductory and “RADIOGENOMICS WORKFLOW“ parts. It is clear for the reader that different people wrote different parts, since possibly different co-authors felt the need of providing the same definition. This should not be perceived in a manuscript. 

Author response: We thank the reviewer for this comment. We have revised the definitions included in the manuscript and harmonized the different parts.

Reviewer response: There are still other sections of the manuscript needing this revision. E.g.Authors revised page 6 and 17, but lots of repeated texts are seen. 

Author response: We thank the reviewer for this comment. We have revised the definitions included in the manuscript, removing the repeated texts. You can find it in yellow on pages 2,3,4,6,7,8,9,10,11,14,16,20,21,23.

  • The authors only focus on MRI scans. However, the scope of the concept of radiomics is broader, as medical images from CT, PET, SPECT, etc. can also be used. This should be addressed by the authors in the introduction before focusing only on MRI scans. 

Author response: We thank the reviewer for this comment. We have explained that this review is focused on MRI radiogenomic approach both in the abstract and in the introduction. You can find it in yellow on pages 2 and 8.

Reviewer response: Pages 8 does not refer this.

Author response: We thank the reviewer for this comment. We have explained that this review is focused on MRI radiogenomic approach both in the abstract and in the introduction. You can find it in yellow on pages 1 and 3.

  • The small conclusions provided by the authors at the end of each topic are a very positive point to highlight. However, it would be great if authors could provide a summary image with a clear association between predictive factor (e.g. IDH mutational status, MGMT promoter methylation status, etc..) and the respective most promissing radiomic features reviewed.

Author response: We thank the reviewer for this comment.  We have discussed common findings and discrepancies of the studies regarding the status of IDH mutation and the status of MGMT promoter methylation, summarizing the most relevant findings. You can find it in yellow on pages 28, 29 and 33.

Reviewer response: Pages 28, 29 and 33 are references. The texts added throughout the text are very confused, repetitive…please clarify if the intention was to put it using bullet points.

Author response: We thank the reviewer for this comment. At the end of the paragraphs named “Prediction of IDH mutational status” and “Predicition of MGMT methylation status” we have provided small conclusions to summarize common findings across different studies. The important premise, reiterated in the text, is that these studies are all retrospective and very heterogeneous, therefore it is very difficult to draw conclusions. Nevertheless, we have tried to summarize the commonalities between the different studies through an easy-to-read bullet list. You can find it in green on pages 16,17 and 21. Moreover, as requested by the reviewer, we have added a figure (figure 3) to make these points easier to understand also from a visual point of view.

Round 3

Reviewer 2 Report

The authors made proper alterations and revisions.

I just highlight one incongruity: Figure 3 mentioned in lines 669 and 842 does not appear in the document.

This review is now worth of publication in Biomedicines with minor reviews.

Author Response

Dear Reviewer, thank you for your comment. 

You can found the figure 3 in the text.